# Conversational Semantic Parsing using Dynamic Context Graphs

**Parag Jain**     **Mirella Lapata**
Institute for Language, Cognition and Computation
School of Informatics, University of Edinburgh
10 Crichton Street, Edinburgh EH8 9AB
parag.jain@ed.ac.uk   mlap@inf.ed.ac.uk

## Abstract

In this paper we consider the task of conversational semantic parsing over general purpose knowledge graphs (KGs) with millions of entities, and thousands of relation-types. We focus on models which are capable of interactively mapping user utterances into executable logical forms (e.g., SPARQL) in the context of the conversational history. Our key idea is to represent information about an utterance and its context via a subgraph which is created dynamically, i.e., the number of nodes varies per utterance. Rather than treating the subgraph as a sequence, we exploit its underlying structure and encode it with a graph neural network which further allows us to represent a large number of (unseen) nodes. Experimental results show that dynamic context modeling is superior to static approaches, delivering performance improvements across the board (i.e., for simple and complex questions). Our results further confirm that modeling the structure of context is better at processing discourse information, (i.e., at handling ellipsis and resolving coreference) and longer interactions.

## 1 Introduction

General purpose knowledge graphs (KG), like Wikidata (Vrandečić and Krötzsch, 2014) structure information in a semantic network of entities, attributes, and relationships, allowing machines to tap into a vast knowledge base of facts. Knowledge base question answering (KBQA) is the task of retrieving answers from a KG, given natural language questions. A popular approach to KBQA (see Gu et al. 2022 and the references therein) is based on semantic parsing which maps questions to logical form queries (e.g., in SPARQL) that return an answer once executed against the KG.

Existing work (e.g., Bogin et al. 2019; Ravishankar et al. 2021; Yin et al. 2021) has mostly focused on answering questions in isolation, whereas we consider the less studied task of *conversational*

---

1. Who starred in Mathias Kneissl ?
   `SELECT ?x WHERE { wd:Q3298576 wdt:P161 ?x . ?x wdt:P31 wd:Q502895 . }`
   Rainer Werner Fassbinder, Volker Schlöndorff, Hanna Schygulla

2. Who was the director of that work of art ?
   `SELECT ?x WHERE { wd:Q3298576 wdt:P57 ?x . ?x wdt:P31 wd:Q502895 . }`
   Reinhard Hauff

3. Does Dubashi have that person as actor ?
   `ASK { wd:Q76025 wdt:P161 wd:Q24807818 . }`
   No

4. Which works of art are Rainer Werner Fassbinder or Laura Esquivel a screenwriter of ?
   `SELECT ?x WHERE { { ?x wdt:P58 wd:Q44426 . ?x wdt:P31 wd:Q838948 . }`
   `UNION { ?x wdt:P58 wd:Q230586 . ?x wdt:P31 wd:Q838948 . } }`
   The American Soldier, Lili Marleen, Love Is Colder Than Death ...

   `Q3298576: Mathias Kneissl, Q76025: Reinhard Hauff, Q24807818: Dubashi, Q44426: Rainer Werner Fassbinder Q230586: Laura Esquivel, Q838948: work of art, Q502895: common name, P161: cast member, P31: instance of, P57: director, P58: screenwriter`

Figure 1: Example interaction from SPICE dataset (Perez-Beltrachini et al., 2023) with utterances, corresponding SPARQL queries, and answers returned after executing the queries on the Wikidata graph engine. The bottom block shows the KG elements (i.e., graph nodes) involved in this interaction.

semantic parsing. Specifically, our interest lies in building systems capable of interactively mapping user utterances to executable logical forms in the *context* of previous utterances. Figure 1 shows an example of a user-system interaction, taken from SPICE (Perez-Beltrachini et al., 2023), a recently released conversational semantic parsing dataset. Each interaction consists of a series of utterances that form a coherent discourse and are translated to executable semantic parses (in this case SPARQL queries). Interpreting each utterance, and mapping it to the correct parse needs to be situated in a particular context as the exchange proceeds. To answer the question in utterance 2, the system needs

to recall that *Mathias Kneissl* is still the subject of the conversation, however, the user is no longer interested in who starred in the film but in who directed it. It is also natural for users to omit previously mentioned information (e.g., through ellipsis or coreference), which would have to be resolved to obtain a complete semantic parse.

In addition to challenges arising from processing contextual information, the semantic parsing task itself involves linking entities, types, and predicates to elements in the KG (e.g., *Mathias Kneissl* to Q3298576) whose topology is often complex with a large number of nodes. Moreover, unlike relational databases, the schema of an entity is not static but dynamically instantiated (Gu et al., 2022). For example, the entity type person can have hundreds of relations but only a fraction of these will be relevant for a specific utterance. Therefore, to generate faithful queries, we cannot rely on memorization and should instead make use of local schema instantiation. In general, narrowing down the set of entities and relations is critical to parsing utterances requiring complex reasoning (i.e., where numerical and logical operators apply over sets of entities).

Existing work (Perez-Beltrachini et al., 2023) handles the aforementioned challenges by adopting various simplifications and shortcuts. For instance, since it is not feasible to encode the entire KG, only a subgraph relevant to the current utterance is extracted and subsequently linearized and treated as a sequence. Entity type information that is not directly accessible via neighboring subgraphs is obtained through a *global* lookup (essentially a reverse index of all types in the KG). This solution is computationally expensive, as the lookup is performed practically for every user utterance, and does not scale well (the index would have to be recreated every time the KG changed).

In this paper we propose a modeling approach to conversational semantic parsing which relies on *dynamic context graphs*. Our key idea is to represent information about an utterance and its context through a dynamically generated subgraph, wherein the number of nodes varies for each utterance. Moreover, rather than treating the subgraph as a sequence, we exploit its underlying structure and encode it with a graph neural network (Scarselli et al., 2009; Gori et al., 2005). To improve generalization, we learn *implicit* node embeddings by aggregating information from neighboring nodes whose embeddings are in turned initialized

through a pretrained model (Devlin et al., 2019). In addition, we introduce context-dependent type linking, based on the entity and its surrounding context which further helps with type disambiguation.

Experimental evaluation on the $\mathbb{SPICE}$ dataset (Perez-Beltrachini et al., 2023) demonstrates that modeling context dynamically is superior to static approaches, improving performance across the board (i.e., for simple and complex questions requiring comparative or quantitative reasoning). Our results further confirm that modeling the structure of context is better at processing discourse information, (i.e., at handling ellipsis and resolving coreference) and longer interactions with multiple turns.

## 2 Related Work

Previous work on semantic parsing for KBQA (Gu et al., 2022) has focused on mapping stand-alone utterances to logical form queries. Various approaches have been proposed to this effect which broadly follow three modeling paradigms. Ranking methods first enumerate candidate queries from the KB and then select the query most similar to the utterance as the semantic parse (Ravishankar et al., 2021; Hu et al., 2021; Bhutani et al., 2019). Coarse-to-fine methods (Dong and Lapata, 2018; Ding et al., 2019; Ravishankar et al., 2021) perform semantic parsing in two stages, by first predicting a query sketch, and then filling in missing details. Finally, generation methods (Yin et al., 2021) first rank candidate parses and then predict the final parse by conditioning on the utterance and best retrieved logical forms.

Our task is most related to conversational text-to-SQL parsing, as manifested in datasets like SParC (Yu et al., 2019b), CoSQL (Yu et al., 2019a), and ATIS (Dahl et al., 1994; Suhr et al., 2018). SParC and CoSQL cover multiple domains and include multi-turn user and system interactions. These datasets are challenging in requiring generalization to unseen databases, but the conversation length is fairly short and the databases relatively small-scale. ATIS contains utterances paired with SQL queries pertaining to a US flight booking task; it exemplifies several long-range discourse phenomena (Jain and Lapata, 2021), however, it covers a single domain with a simple database schema.

Graph-based methods have been previously employed in semantic parsing primarily to encode the database schema, so as to enable the parser to globally reason about the structure of the output query

(Bogin et al., 2019). Other work (Hui et al., 2022) uses relational graph networks to jointly represent the database schema and syntactic dependency information from the questions. In the context of conversational semantic parsing, Cai and Wan (2020) use a graph encoder to model how the elements of the database schema interact with information in preceding context. Along similar lines, Hui et al. (2021), use a graph neural network with a dynamic memory decay mechanism to model the interaction of the database schema and the utterances in context as the conversation proceeds. All these approaches encode the schema of relational databases which are significantly smaller in size (e.g., number of entities and types of relations) compared to large-scale KGs, where encoding the entire graph in memory is not feasible.

Closest to our work are methods which cast conversational KGQA as a semantic parsing task (Kacupaj et al., 2021; Marion et al., 2021). These approaches build hand-crafted grammars that are not directly executable to a KG engine. Furthermore, they assume the KG can be fully encoded in memory which may not be feasible in real-world settings. Perez-Beltrachini et al. (2023) develop a parser which is executable with a real KG engine (e.g., Blazegraph) but simplify the task by considering only limited conversation context.

## 3 Problem Formulation

Given a general purpose knowledge graph, such as Wikidata, our task is to map user utterances to formal executable queries, SPARQL in our case. We further assume an *interactive* setting where users converse with the system in natural language and the system responds while taking into account what has already been said (see Figure 1). The system's response is obtained upon *executing* the query against a graph query engine.

Let $G$ denote the underlying KG and $I$ a single interaction. $I$ consists of a sequence of turns where each turn is represented by $\langle X_t, A_t, Y_t \rangle$ denoting an utterance-answer-query triple at time $t$ (see blocks in Figure 1). A user utterance $X_t$ is a sequence of tokens $\langle x_1, x_2, \ldots, x_{|X_t|} \rangle$, where $|X_t|$ is the length of the sequence and each $x_i, i \in [1, |X_t|]$ is a natural language token. Query string $Y_t$ is a sequence of tokens $\langle y_1, y_2, \ldots, y_{|Y_t|} \rangle$, where $|Y_t|$ is the length of the sequence and each $y_i, i \in [1, |Y_t|]$ is a either a token from the SPARQL syntax vocabulary (e.g., SELECT, WHERE) or a KG element $\in G$

(e.g., Q3298576). Answer $A_t$ at time $t$ is the result of executing $Y_t$ against $G$. Given the interaction history $I[: t-1]$ at turn $t$ and current utterance $X_t$, our goal is to generate $Y_t$. This involves understanding $X_t$ in the context of $X_{:t-1}$, $A_{:t-1}$, and $G$, and learning to generate $Y_t$ based on encoded contextual information.

## 4 Model

Our modeling approach combines three components. We first ground named entities in the user utterance to KG entities, and use these linked entities to extract a subgraph that functions as context (Section 4.1). The second component is responsible for type linking in the context of current and previously mentioned named entities (Section 4.2). And finally, our semantic parser learns to map user utterances into SPARQL queries (Section 4.3).

### 4.1 Entity Grounding and Disambiguation

We are interested in grounding user utterance $X_t$ to graph $G$. Since encoding the entire KG is not feasible, we extract a subgraph from $G$ which is relevant to the current turn. To achieve this, we first perform named entity recognition with an off-the-shelf NER system, in our experiments we use AllenNLP (Gardner et al., 2018). We perform named entity linking through efficient string matching[1] (Aho and Corasick, 1975) unlike Perez-Beltrachini et al. (2023) who deploy an ElasticSearch server for querying an inverted index.

A string can be ambiguous, i.e., link to multiple entities. For example, *Rainer Werner Fassbinder* can be linked to *filmmaker* (Q44426) and *movie* (Q33561976). To deal with ambiguity and to increase recall, Perez-Beltrachini et al. (2023) do not ever commit to a single entity but instead include the top-$K$ matching ones; however, this introduces noise and increased computational cost. Instead, we disambiguate entities based on their popularity in the training set (Shen et al., 2015) and compare the two approaches in Section 6.

Following Perez-Beltrachini et al. (2023), for each identified KG entity $e$, we extract triples $(e, r, o)$ and $(s, r, e)$, where $s$ and $o$ denote the subject or object of relation $r$. For instance, entity *Dubashi* would have triple *(Dubashi, country of origin, India)*. Subjects and objects are further mapped to their types in place of actual entities

---

[1] Our implementation is based on pyahocorasick https://pypi.org/project/pyahocorasick/.

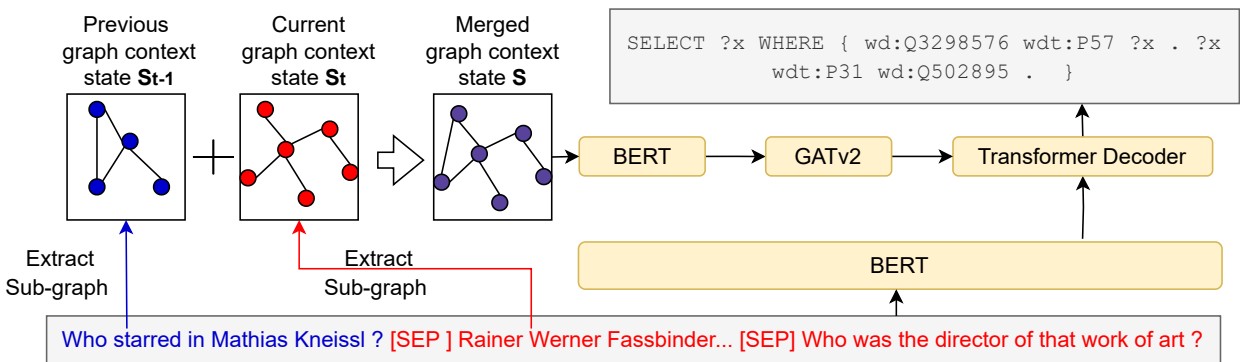

Figure 2: Model architecture. The previous and current utterance are concatenated and their subgraphs are merged and encoded in a graph neural network. The subgraphs represent the entity neighborhood and type linking.

(i.e., $(e, r, o_{type})$ and $(s_{type}, r, e)$). In our example, the triple for *Dubashi* then becomes *(Dubashi, country of origin, country)*, where *country* is type Q6256. We denote the set of typed triples as $G_t^{ent}$.

## 4.2 Context-Dependent Type Linking

Entities in SPARQL queries have types, for example, in Figure 1, KG element Q502895 is a placeholder for the type "common name". Type instances are often present in the one-hop entity neighborhood $G_t^{ent}$, but can also be more hops away. Perez-Beltrachini et al. (2023) index *all* KG types and perform a *global* lookup which is computationally expensive, and solely applicable to the KG they are working with. Instead, we perform type linking based on the entities mentioned in the *current* context. We expand the grounded entities to extract triples with type information.[2] Since considering multi-hop neighborhoods would lead to extremely large subgraphs and would not be memory efficient, we prune these based on their string overlap with the user utterance, significantly reducing the number of triples. The pruned graph $G_t^{type}$ is merged with the previously obtained entity graph $G_t^{ent}$, such that, $G_t = G_t^{ent} \cup G_t^{type}$.

## 4.3 Dynamic Context Graph Model

Figure 2 shows the overall architecture of our dynamic context graph model (which we abbreviate to DCG). DCG takes as input a tuple of form $\langle C_t, X_t, G_t \rangle$, where $X_t$ is a user utterance at time $t$ and $G_t$ is the corresponding subgraph. $C_t$ denotes the previous context information that includes the previous utterance $X_{t-1}$, the previous answer

$A_{t-1}$, and subgraph $G_{t-1}$. We use $\hat{G}_t$ to represent merged context subgraphs $G_t$ and $G_{t-1}$ such that $\hat{G}_t = G_t \cup G_{t-1}$. We encode the context subgraph $\hat{G}_t$ with a graph neural network (GNN, Scarselli et al. 2009; Gori et al. 2005) and user utterances and their discourse context with BERT (Devlin et al., 2019). Our decoder is a transformer network (Vaswani et al., 2017) that conditions on the user utterance encoding and the corresponding graph representations.

**Utterance Encoder** We use BERT[3] (Devlin et al., 2019) to represent the concatenation of previous utterance $X_{t-1}$, previous answer $A_{t-1}$, and current utterance $X_t$ (see Figure 2). To distinguish between current and past context we use the [SEP] token. More formally, let $\hat{X}_t = $ [CLS] $X_{t-1}$ [SEP] $A_{t-1}$ [SEP] $X_t$ denote the input to BERT, where $X_t = (x_1, x_2 \dots x_{|X_t|})$, $A_{t-1} = (a_1, a_2 \dots a_{|A_{t-1}|})$, and $X_{t-1} = (x_1, x_2 \dots x_{|X_{t-1}|})$ are sequences of natural language tokens. We obtain latent representations $Z_t$ as $Z_t = \text{BERT}(\hat{X}_t)$.

**Graph Encoder** We represent the KG subgraph $\hat{G}_t$ at time $t$ as a directed graph $\mathcal{G} = (\mathcal{V}, \mathcal{E})$ (hereafter, we simplify notation and drop time $t$), where $\mathcal{V} = \{v_1, v_2, \dots, v_n\}$ are nodes such that $v_i \in \{entities, relations, types\}$ and $\mathcal{E} \subseteq \mathcal{V} \times \mathcal{V}$. Each node $v_i$ consists of a sequence of natural language tokens, such that $v_i = \langle v_{i1}, v_{i2}, \dots, v_{i|v_i|} \rangle$. Our KG has a large number of distinct nodes, but we cannot possibly attest all of them during training. To handle unseen nodes at test time, we obtain a generic node representa-

---

[2]For entity *ent* we query `select ?r1 ?n1 ?t1 ?r2 ?n2 ?t2 where { wd:ent ?r1 ?n1 . ?n1 wdt:P31 ?t1 . OPTIONAL {?n1 ?r2 ?n2 . ?n2 wdt:P31 ?t2} }`.

[3]We employ BERT for a fair comparison with prior work. Nonetheless, our model does not have any inherent restrictions that would prevent the use of other pretrained models.

tion $h_i^0$ for node $v_i$, where $h_i^0 = \text{AVG}(\text{BERT}(v_i))$. In other words, we compute encoding $h_i^0$ by taking the average of the individual token encodings obtained from BERT. We do not create a separate embedding matrix but directly update the BERT representations during learning, which allows us to scale to a large number of (unseen) nodes.

A graph neural network (GNN) learns node representations by aggregating information from its neighboring nodes. Each GNN layer $l$, takes as input node representations $\{h_i^{l-1} \mid i \in [1, n]\}$ and edges $\mathcal{E}$. The output of each layer is an updated set of node representations $\{h_i^l \mid i \in [1, n]\}$. We use Graph Attention Network v2 (GATv2, Brody et al. 2022) for updating node representations which replaces the static attention mechanism of GAT (Velickovic et al., 2018) with a dynamic and more expressive variant. Let $\mathcal{N}_i = \{v_j \in \mathcal{V} \mid (j, i) \in \mathcal{E}\}$ denote the neighbors of node $v_i$ and $\alpha_{ij}$ the attention score between node $h_i$ and $h_j$. We calculate attention as a single-layer feedforward neural network, parametrized by a weight vector $a$ and weight matrix $W$:

$$\alpha_{ij} = \frac{\exp\big(\psi\big(h_i^{l-1}, h_j^{l-1}\big)\big)}{\sum_{k \in \mathcal{N}_i} \exp\big(\psi\big(h_i^{l-1}, h_k^{l-1}\big)\big)} \quad (1)$$

The scoring function $\psi$ is computed as follows:

$$\psi\big(h_i^{l-1}, h_j^{l-1}\big) = \\ a^T \text{LeakyReLU}\big(W \cdot [h_i^{l-1} \parallel h_j^{l-1}]\big) \quad (2)$$

where $\cdot^T$ represents transposition and $\parallel$ is the concatenation operation. Attention coefficients corresponding to each node $i$ are then used to compute a linear combination of the features corresponding to neighboring nodes as:

$$h_i^l = \sigma\left(\sum_{j \in \mathcal{N}_i} \alpha_{ij} W h_j^{l-1}\right) \quad (3)$$

**Decoder** Our decoder is a transformer network (Vaswani et al., 2017). Let $H_t^l = \big(h_{1t}^l, h_{2t}^l, \ldots, h_{nt}^l\big)$ denote the sequence of node representations from the last layer of the graph network (recall $t$ here represents an interaction turn). Our decoder models the probability of generating a SPARQL parse conditioned on the graph and input context representations, i.e., $p(Y_t \mid H_t^l, Z_t)$. Generating the SPARQL parse requires generating syntax symbols (such as SELECT, WHERE) and KG elements (i.e., entities, types, and relations). Given

decoder state $s_i$ at the $i^{th}$ generation step, the probability of generating $y_i$ is calculated as:

$$p(y_i \mid y_{<i}, H_t^l, s_i) = p_{\mathcal{G}}(y_i \mid y_{<i}, H_t^l, s_i) \\ + p_{\mathcal{S}}(y_i \mid y_{<i}, s_i) \quad (4)$$

where $p_{\mathcal{G}}$ and $p_{\mathcal{S}}$ are the probability of generating a graph node and syntax symbol, respectively. We calculate $p_{\mathcal{S}} = \text{softmax}(W_1 s_i)$, such that $W_1 \in \mathbb{R}^{|V_s| \times d}$, and $|V_s|$ is the SPARQL syntax vocabulary size. We calculate $p_{\mathcal{G}}$ using node embeddings $H_t^l$, as $p_{\mathcal{G}} = \text{softmax}(H_t^l s_i)$.

**Training** Our model is trained end-to-end by optimizing the cross-entropy loss. Given training instance $\langle C_t, X_t, Y_t, G_t \rangle$, where $Y_t$ is a sequence of gold output tokens $\langle y_1, y_2, \ldots, y_{|Y_t|} \rangle$, we minimize the token-level cross-entropy as:

$$\mathcal{L}(\hat{y}_i) = -log\, p(y_i \mid X_t, G_t, C_t) \quad (5)$$

where $\hat{y}_i$ denotes the predicted output token at decoder step $i$.

## 5 Experimental Setup

**Dataset** We performed experiments on $\mathbb{SPICE}$ (Perez-Beltrachini et al., 2023), a recently released large-scale dataset[4] for conversational semantic parsing built on top of the CSQA benchmark (Saha et al., 2018). $\mathbb{SPICE}$ consists of user-system interactions where natural language questions are paired with SPARQL parses and answers provided by the system correspond to SPARQL execution results (see Figure 1). We present summary statistics of the dataset in Table 1. As can be seen, it contains a large number of training instances, the conversations are relatively long (the average turn length is 9.5), and the underlying KG is sizeable (12.8M entities). $\mathbb{SPICE}$ has simple factual questions but also more complicated ones requiring reasoning over sets of triples; it also exemplifies various discourse-related phenomena such as coreference and ellipsis. We provide examples of the types of questions attested in $\mathbb{SPICE}$ in Appendix C.

**Evaluation Metrics** Following previous work (Perez-Beltrachini et al., 2023), we report exact match accuracy and F1 (or accuracy depending on question type). Exact match is the percentage of predicted SPARQL queries that string match with the corresponding gold SPARQL. F1 (or

---

[4] https://github.com/EdinburghNLP/SPICE.

| | |
|---|---|
| # instances | 197K |
| # entities | 12.8M |
| # relations | 2,738 |
| # types | 3,064 |
| Avg. turn length | 9.5 |
| Avg. entities per conversation | 7.6 |
| Avg. types per conversations | 6.5 |
| Avg. neighborhood per turn | 181.4 triples |

Table 1: Statistics of the $\mathbb{SPICE}$ dataset.

answer accuracy) is calculated between execution results of predicted queries and gold queries. For Verification queries and queries involving Quantitative and Comparative Reasoning, we calculate execution answer accuracy. For other types of questions, F1 scores are calculated by treating the results as a set.

**Model Configuration** Our model is implemented using PyTorch (Paszke et al., 2019) and trained with the AdamW (Loshchilov and Hutter, 2019) optimizer.[5] Model selection was based on exact match accuracy on the validation set. We used two decoder layers and two GATv2 layers for all experiments. We used HuggingFace's pretrained BERT embeddings (Wolf et al., 2020), specifically the uncased base version. Our GATv2 implementation is based on PyTorch Geometric (Fey and Lenssen, 2019) with two attentions heads. We use adjacency matrices stacking as a method of creating mini-batches for our GNN across different examples. We identify named entities using the AllenNLP named entity recognition (NER) system (Gardner et al., 2018). Our execution results are based on the Wikidata subgraph provided by Perez-Beltrachini et al. (2023). Our SPARQL server is deployed using Blazegraph.[6] See Appendix B for more implementation details.

As described in Section 4.3, at each utterance, our model encodes the previous $t$ subgraphs. Larger context is informative but can also introduce noise. We treat $t$ as a hyperparameter and optimize its value on the development set. We report results with $t = 5$ (see Appendix A for details).

**Comparison Models** We compare against the semantic parser of Perez-Beltrachini et al. (2023). Their model is based on BERT (Devlin et al., 2019), it relies on AllenNLP (Gardner et al., 2018) for named entity recognition, and performs entity link-

ing with an ElasticSearch[7] inverted index. As mentioned earlier, they do not explicitly perform named entity disambiguation (they consider the $K = 5$ best matching entities and their neighborhood graphs as part of the vocabulary) and use a global lookup for type linking. As the size of the linearized subgraph often exceeds BERT's maximum number of input tokens (which is 512), they adopt a workaround where the graph is chunked into several subsequences, and encoded separately. We refer to their Semantic Parser as BertSP$_{\mathcal{GL}}$, where $\mathcal{GL}$ is a shorthand for Global Lookup.

In addition to our full dynamic context graph model which performs Context-dependent Type Linking (DCG$_{\mathcal{CL}}$), we also build a simpler variant (DCG) which only relies on the entity neighborhood subgraph for type information. Moreover, we create two variants of our model, one which disambiguates entities, and another one which does not (similar to Perez-Beltrachini et al. 2023).

## 6 Results

In this section, we evaluate the performance of our semantic parser on the $\mathbb{SPICE}$ test set. We report results on individual question types and overall. We also analyze our system's ability to handle different discourse phenomena like ellipsis and coreference as well as interactions of varying length.

**The Effect of Dynamic Context** Table 2 summarizes our results. We first concentrate on model variants *without entity disambiguation* for a fair comparison with Perez-Beltrachini et al. (2023).

We compare DCG$_{\mathcal{GL}}$, a version of our model which adopts a global lookup for type liking similar to BertSP$_{\mathcal{GL}}$ and differs only in how contextual information is encoded. As we can see, our graph-based model performs better, reaching an F1 score of 72.3% compared to 59% obtained by BertSP$_{\mathcal{GL}}$ which is limited by the way it encodes contextual information. Recall, that BertSP$_{\mathcal{GL}}$ linearizes the subgraph context, splits into subsequences and feeds it to the model chunk-by-chunk. Our model alleviates this problem by efficiently encoding the KG information with a graph neural network, preserving dependencies captured in its structure. As a result, DCG$_{\mathcal{GL}}$ performs better on most question types, including simple questions, and reasoning-style questions. We further compare DCG$_{\mathcal{GL}}$ against a variant which uses context-dependent type linking

---

[5]Our code can be downloaded from `https://github.com/parajain/dynamic_context`.

[6]`https://blazegraph.com/`

[7]`https://www.elastic.co/`

| | Without Disambiguation | | | | | | With Disambiguation | | | |
| | DCG$_{\mathcal{CL}}$ | | DCG$_{\mathcal{GL}}$ | | BertSP$_{\mathcal{GL}}$ | | DCG$_{\mathcal{CL}}$ | | DCG | |
| Question Type | F1 | EM | F1 | EM | F1 | EM | F1 | EM | F1 | EM |
|---|---|---|---|---|---|---|---|---|---|---|
| Clarification | 78.60 | 73.63 | 80.42 | 69.47 | **83.91** | 76.58 | 82.01 | 74.82 | 82.03 | 72.10 |
| Logical Reasoning | 64.12 | 49.51 | 51.14 | 31.54 | 22.74 | 28.61 | **93.95** | 79.52 | 93.33 | 78.19 |
| Quantitative Reasoning | 55.66 | 26.29 | **93.25** | 76.88 | 76.20 | 59.01 | 59.83 | 31.17 | 56.66 | 28.66 |
| Comparative Reasoning | 76.06 | 35.59 | 80.59 | 47.28 | 69.56 | 39.37 | **90.91** | 62.46 | 90.09 | 61.11 |
| Simple Question (Coref) | 86.36 | 72.03 | 84.92 | 67.15 | 76.51 | 58.83 | **88.49** | 79.90 | 87.41 | 79.18 |
| Simple Question (Direct) | 87.29 | 71.10 | 83.24 | 65.89 | 71.43 | 58.71 | **88.27** | 62.25 | 85.60 | 61.44 |
| Simple Question (Ellipsis) | 65.22 | 56.08 | 52.84 | 47.48 | 58.14 | 50.90 | 79.08 | 83.87 | **84.35** | 82.45 |
| | AC | EM | AC | EM | AC | EM | AC | EM | AC | EM |
| Verification (Boolean) | 78.30 | 36.97 | 69.07 | 21.00 | 37.16 | 24.90 | **87.41** | 66.32 | 86.75 | 63.66 |
| Quantitative Reasoning (Count) | 61.10 | 56.94 | 66.59 | 62.70 | 50.86 | 48.44 | **75.20** | 70.84 | 72.96 | 69.02 |
| Comparative Reasoning (Count) | 42.79 | 30.40 | 60.91 | 44.68 | 43.48 | 40.67 | **67.70** | 57.34 | 66.76 | 56.60 |
| Overall | 69.55 | 50.85 | 72.30 | 53.41 | 59.00 | 48.60 | **81.28** | 66.85 | 80.59 | 65.24 |

Table 2: Results on $\mathbb{SPICE}$ dataset (test set). BertSP$_{\mathcal{GL}}$ (Perez-Beltrachini et al., 2023) uses NER based on AllenNLP) and global look-up (subscript $_{\mathcal{GL}}$) for type linking. DCG$_{\mathcal{CL}}$ uses context-dependent type linking (subscript $_{\mathcal{CL}}$) and also AllenNLP. DCG has no type linking. We measure F1, Accuracy (AC), and Exact Match (EM).

instead (DCG$_{\mathcal{CL}}$). We find that context-dependent type linking is slightly worse than global lookup which is expected given that it does not have access to the full list of KG types.

In general, we observe that results with exact match (EM) are lower than F1 or Accuracy. EM is a stricter metric, it does not allow for any deviation from the goldstandard $\mathrm{SPARQL}$. However, it is possible for two queries to have different syntax but equivalent meaning, and for partially well-formed queries to evaluate to partially accurate results. In contrast to EM, F1 and Accuracy give partial credit and thus obtain higher scores.

**The Effect of Entity Disambiguation** We now present results with a variant of our model which operates over *disambiguated entities* (compare second and fifth blocks in Table 2, with heading DCG$_{\mathcal{CL}}$). We observe that disambiguation has a significant effect on model performance, leading to an F1 increase of more than 11%. We further assess the utility of context-dependent linking by comparing DCG$_{\mathcal{CL}}$ to a variant which does not have access to the type graph $G_t^{type}$, neither during training nor during evaluation (see column DCG, With Disambiguation). This type-deficient model performs overall worse both in terms of F1 and EM, but is still superior to BertSP$_{\mathcal{GL}}$, even though the latter has access to more information via the top-$K$ entity lookup and global type linking. This points to the importance of encoding context in a targeted manner rather than brute force. In Appendix A we discuss the effect of context length on the availability of type information.

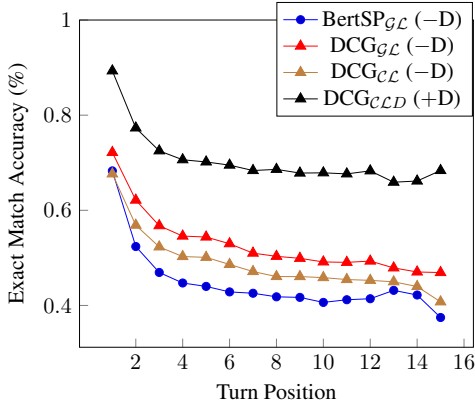

Figure 3: Exact Match Accuracy averaged across question types at different turn positions. $+/-$D denotes the presence/absence of entity disambiguation.

**The Effect of Conversation Length** We next examine the benefits of modeling context dynamically. Ideally, a model should produce an accurate semantic parse no matter the conversation length. Figure 3 plots exact match accuracy (averaged across question types) against different turn positions. In general, we observe that utterances occurring later in the conversation are more difficult to parse. As the dialogue progresses, subsequent turns become more challenging for the model which is expected to leverage the common ground established so far. This involves maintaining the subgraph context based on the conversation history in addition to handling linguistic phenomena such as coreference and ellipsis. Overall, DCG$_{\mathcal{CL}}$ (with and without entity disambiguation) is superior to BertSP$_{\mathcal{GL}}$, and the gap between the two models is more pronounced for later turns.

|            | Without Disambiguation | | | With Disambiguation | |
|------------|------------------|------------------|------------------|------------------|--------|
|            | $DCG_{CL}$ | $DCG_{GL}$ | $BertSP_{GL}$ | $DCG_{CL}$ | $DCG$ |
| Coref$=-1$ | 58.25 | 51.10 | 49.39 | **74.23** | 72.22 |
| Coref$< -1$ | 23.52 | 12.42 | 00.00 | **33.64** | 32.40 |
| Ellipsis | 39.30 | 39.90 | 26.39 | **62.26** | 61.10 |
| MEntities | 43.71 | 53.46 | 41.64 | **61.59** | 58.90 |

Table 3: Average Exact Match accuracy. Coref$=-1$ are utterances with referring expressions resolved in the previous turn. Coref$< -1$ are utterances with referring expressions resolved in the wider discourse context, beyond the previous turn. MEntities abbreviates multiple entities and refers to utterances with plural mentions.

**Modeling Discourse Phenomena**  Discourse phenomena, such as ellipsis and coreference, are prevalent in conversations. Ellipsis refers to grammatical omissions from an utterance that can be recovered from context. In the interaction:

(Q1)    What does Andrei Neagoe do for a living?

(Q2)    And how about Wilhelm Dietrichson?

the phrase *do for a living* is elided from (Q2) but can be understood in the context of (Q1). Coreference on the other hand, occurs between utterances that refer to the same entity. For example, between utterances 2 and 3 in Figure 1.

In Table 3, using exact match, we assess how different models handle ellipsis and coreference across question types. Coref$=-1$ refers to cases where coreference can be resolved in the immediate context, i.e., the previous turn. Coref$< -1$ involves utterances that require access to wider conversation context, beyond the previous turn. In the setting that does not disambiguate entities, we observe that models which exploit discourse context (variants $DCG_{GL}$ and $DCG_{CL}$) are better at resolving co-referring and elliptical expressions compared to $BertSP_{GL}$. We also see that entity disambiguation is very helpful, leading to substantial improvements for $DCG_{CL}$ across discourse-related phenomena.

Similar to Perez-Beltrachini et al. (2023), we also evaluate model performance on utterances with plural mentions; these are typically linked to multiple entities which the semantic parser must enumerate in order to build a correct parse (MEntities in Table 3). $DCG_{CL}$ with disambiguation is overall best, while DCG (without type linking) is worse. This is not surprising, utterances with multiple entities generally have complex parses, with multiple sub-queries and entity types, which DCG does not have access to.

**The Nature of Parsing Errors**  Overall, we find that our model is able to predict syntactically valid SPARQL queries. Errors are mostly due to misinterpretations of the question's intent given the graph context and previous questions or missing information. Our model also has difficulty parsing Clarification and Quantitative Reasoning questions. For Clarification questions, it is not able to select the right entity after clarification. For example, in the following conversation:

| Answer: | Peter G. Van Winkle, Arthur I. Boreman, William E. Stevenson |
|---|---|
| Utterance: | Which language is that person capable of writing ? |
| Clarification: | Did you mean Arthur I. Boreman ? |
| Answer: | No, I meant Peter G. Van Winkle. Could you tell me the answer for that? |

it selects *Arthur I. Boreman* (Q709961) instead of *Peter G. Van Winkle* (Q1404201) leading to an incorrect SQL parse. In this case, the broader context overrides useful information in the immediately preceding turn. Determining relevant context based on specific question intents would be helpful, however, we leave this to future work.

Failures in type linking are a major cause of errors for Quantitative Reasoning questions which typically have no or very limited context (e.g., "Which railway stations were managed by exactly 1 social group ?"). However, our model relies on the availability of types in the entity neighborhood, as it performs type linking in a context dependent manner. We observe that it becomes better at parsing such questions when given access to all KG types (see Table 2, $DCG_{GL}$ vs. $DCG_{CL}$).

## 7   Conclusions

In this paper, we present a semantic parser for KBQA which interactively maps user utterances into executable logical forms, in the context of previous utterances. Our model represents information about utterances and their context as KG subgraphs which are created dynamically and encoded using a graph neural network. We further propose a context-dependent approach to type linking which is efficient and scalable.

Our experiments reveal that better modeling of contextual information improves performance, in terms of entity and type linking, resolving coreference and ellipsis, and keeping track of the interaction history as the conversation evolves. Directions for future work are many and varied. In

experiments, we use an off-the shelf NER system, however, jointly learning a semantic parser and an entity linker would be mutually beneficial, avoiding error propagation. Given that it is prohibitive to encode the entire KG, we encode relevant subgraphs on the fly. We could further explicitly model the relationship between KG entities and question tokens which previous work has shown is promising (Wang et al., 2020). Finally, it would be interesting to adapt our model so as to handle non-i.i.d generalization settings.

## 8 Limitations

Our model relies on a pre-trained NER module for entity linking. As this module is trained and evaluated on specific datasets, its performance may not generalize on unseen domains within Wikidata. Moreover, we did not explicitly consider relations. We assume that the correct information will be available which may not always be the case. We focus on encoding KG structural information and pass the learning of the interactions between the KG and the linguistic utterances to the decoder. As shown in previous work (Zhang et al., 2022), effectively combining KG information with a language model can be mutually beneficial in the context of question answering. However, it requires an extensive study in itself to determine the task specific parametrization (Wang et al., 2022).

### Acknowledgements

This work is supported in part by Huawei and the UKRI Centre for Doctoral Training in Natural Language Processing (grant EP/S022481/1). Lapata gratefully acknowledges the support the UK Engineering and Physical Sciences Research Council (grant EP/W002876/1).

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

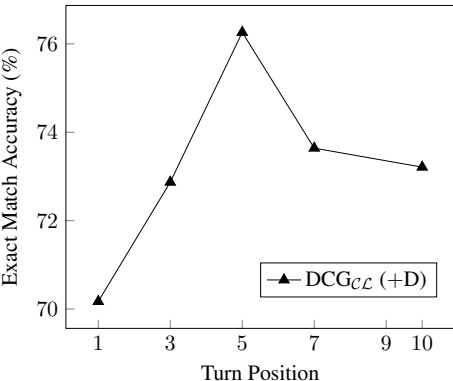

Figure 4: Exact Match Accuracy for different context lengths averaged across question types (validation set; +D: with entity disambiguation).

and Jure Leskovec. 2022. GreaseLM: Graph REASoning enhanced language models. In *The Tenth International Conference on Learning Representations, ICLR 2022, Virtual Event, April 25-29, 2022*. OpenReview.net.

## A The Effect of Context Length

Figure 4 plots the performance of $DCG_{CL}$ (disambiguation setting) against progressively increasing context length $\in [1, 10]$. We observe that access to wider context is beneficial up to a point. Performance deteriorates with very long contexts (beyond turn position 5). We stipulate two reasons for this. Firstly, longer interactions might be long because users ask about more than one entity or topic, in which case local context might be sufficient to provide an answer. And secondly, longer interactions might be genuinely confusing and noisy for annotators to create, let alone models.

We further assess how context length interacts with the availability of type information. Table 4 shows the difference in performance with and without explicit type linking at context lengths 1 and 5. As described in Section 5, DCG does not have explicit type linking while $DCG_{CL}$ uses context-dependent linking, while both models apply entity disambiguation. $\Delta_{F1_1}$ is the absolute difference in F1 score between $DCG_{CL}$ and DCG for context length 1. Similarly, $\Delta_{F1_5}$ denotes the difference for context length 5.

Overall, we find a significant drop in performance for context length 1 compared to context length 5. This indicates that more type information becomes available with increased context length. However, performance varies with question types. Specifically, the exact match difference is lot bigger for Clarification questions compared to Quantita-

| Question Type | Context Length 1 | | | | Context Length 5 | | | | Diff due to type linking | | | |
|---|---|---|---|---|---|---|---|---|---|---|---|---|
| | $\text{DCG}_{\mathcal{CL}}$ | | DCG | | $\text{DCG}_{\mathcal{CL}}$ | | DCG | | | | | |
| | F1 | EM | F1 | EM | F1 | EM | F1 | EM | $\Delta_{\text{F1}_1}$ | $\Delta_{\text{EM}_1}$ | $\Delta_{\text{F1}_5}$ | $\Delta_{\text{EM}_5}$ |
| Clarification | 75.66 | 68.61 | 52.22 | 53.87 | 82.01 | 74.82 | 82.03 | 72.1 | 23.44 | 14.74 | 0.02 | 2.72 |
| Logical Reasoning | 92.59 | 77.07 | 86.9 | 67.94 | 93.95 | 79.52 | 93.33 | 78.19 | 5.69 | 9.13 | 0.62 | 1.33 |
| Quantitative Reasoning | 36.1 | 13.74 | 30.91 | 11.04 | 59.83 | 31.17 | 56.66 | 28.66 | 5.19 | 2.7 | 3.17 | 2.51 |
| Comparative Reasoning | 76.72 | 39.75 | 74.77 | 37.79 | 90.91 | 62.46 | 90.09 | 61.11 | 1.95 | 1.96 | 0.82 | 1.35 |
| Simple Question (Coref) | 88.18 | 79.83 | 83.04 | 76.27 | 88.49 | 79.9 | 87.41 | 79.18 | 5.14 | 3.56 | 1.08 | 0.72 |
| Simple Question (Direct) | 87.56 | 61.59 | 79.74 | 58.21 | 88.27 | 62.25 | 85.6 | 61.44 | 7.82 | 3.38 | 2.67 | 0.81 |
| Simple Question (Ellipsis) | 80.38 | 81.75 | 74.2 | 76.84 | 79.08 | 83.87 | 84.35 | 82.45 | 6.18 | 4.91 | 5.27 | 1.42 |
| | AC | EM | AC | EM | AC | EM | AC | EM | $\Delta_{\text{AC}_1}$ | $\Delta_{\text{EM}_1}$ | $\Delta_{\text{AC}_5}$ | $\Delta_{\text{EM}_5}$ |
| Verification (Boolean) | 88.02 | 61.45 | 82.15 | 48.24 | 87.41 | 66.32 | 86.75 | 63.66 | 5.87 | 13.21 | 0.66 | 2.66 |
| Quantitative Reasoning (Count) | 69.41 | 65.34 | 65.16 | 60.58 | 75.2 | 70.84 | 72.96 | 69.02 | 4.25 | 4.76 | 2.24 | 1.82 |
| Comparative Reasoning (Count) | 42.81 | 30.5 | 39.74 | 28.69 | 67.7 | 57.34 | 66.76 | 56.6 | 3.07 | 1.81 | 0.94 | 0.74 |
| Overall | 73.74 | 57.96 | 66.88 | 51.95 | 81.28 | 66.85 | 80.59 | 65.24 | 6.86 | 6.01 | 0.69 | 1.61 |

Table 4: Interaction of context length and type linking. $\Delta_{\text{F1}_1}$ is the absolute difference in F1 score between $\text{DCG}_{\mathcal{CL}}$ and DCG for context length 1. $\Delta_{\text{F1}_5}$ is the absolute F1 difference for context length 5.

tive Reasoning questions which seem to require access to larger KB subgraphs.

## B  Model Details

Our model is implemented using Py-Torch (Paszke et al., 2019) and trained with the AdamW (Loshchilov and Hutter, 2019) optimizer. It was trained with an A100 GPU with a batch size of 64 and an initial learning rate of 0.001. AdamW coefficients $\beta_1$ and $\beta_2$ (used for computing running averages of gradient and its square) were set to 0.9 and 0.999, respectively. W The weight decay coefficient was set to 0.01 for all experiments. Hyperparameters were set based on initial experiments using a manually selected grid. We did not tune learning rate parameters. We choose the number of GATv2 and decoder layers from $[1, 4]$ and found 2 to work best. Our SPARQL query server was deployed using Blazegraph.[8] which uses only CPU-based resources and has access to 100G of RAM.

We use two attention heads with GATv2. Specifically, let $K$ denote the attention head as computed (Velickovic et al., 2018) in Equation (3). The output of each head is concatenated as follows:

$$h^l = \overset{K}{\underset{k=1}{\big\|}} \sigma \left( \sum_{j \in \mathcal{N}_i} \alpha_{ij}^k W^k h_j^{l-1} \right)$$

where $\|$ represents concatenation. $\alpha_{ij}^k$ are normalized attention coefficients computed by the $k$-th attention mechanism as in Equation (1).

Our graph is represented as an adjacency matrix. To create a mini-batch, adjacency matrices are diagonally stacked (Fey and Lenssen, 2019). This creates a combined graph that holds multiple isolated subgraphs as shown below:

$$\mathbf{A} = \begin{bmatrix} \mathbf{A}_1 & & \\ & \ddots & \\ & & \mathbf{A}_n \end{bmatrix}$$

where $n$ is the batch-size number of graphs. Node input $H$ and target $\bar{H}$ features are simply concatenated in the node dimension as follows:

$$\mathbf{H} = \begin{bmatrix} \mathbf{H}_1 \\ \vdots \\ \mathbf{H}_n \end{bmatrix}, \qquad \bar{\mathbf{H}} = \begin{bmatrix} \bar{\mathbf{H}}_1 \\ \vdots \\ \bar{\mathbf{H}}_n \end{bmatrix}.$$

---

[8] https://blazegraph.com/

## C   The 𝕊ℙ𝕀ℂ𝔼 Dataset: Question Types

| | |
|---|---|
| Simple Question (Ellipsis) | Utterance: Who created the design for Samus Aran?
Answer: Hiroji Kiyotake
Utterance: And how about The Dreamland Chronicles: Freedom Ridge?
Answer: Julian Gollop |
| Simple Question (Direct) | Utterance: Who starred in Mathias Kneissl ?
Answer: Rainer Werner Fassbinder, Volker Schlöndorff, Hanna Schygulla |
| Simple Question (Coreferenced) | Utterance: Who was the director of that work of art ?
Answer: Reinhard Hauff |
| Verification (Boolean) | Utterance: Does Dubashi have that person as actor ?
Answer: No |
| Logical Reasoning (All) | Utterance: Which works of art are Rainer Werner Fassbinder or
Laura Esquivel a screenwriter of ?
Answer: The American Soldier, Lili Marleen, Love Is Colder Than Death... |
| Clarification | Utterance: How many administrative territories or political territories
did that work of art originate ?
Did you mean Lili Marleen ?
No, I meant Querelle. Could you tell me the answer for that?
Answer: 2 |
| Quantitative Reasoning (Count) | Utterance: How many still waters are situated nearby Norway
or Austria-Hungary ?
Answer: 2 |
| Comparative Reasoning (Count) | Utterance: How many watercourses are more number of landscapes
located on than Kafue River ?
Answer: 2 |
| Comparative Reasoning (All) | Utterance: Which political territories are located nearby lesser number
of watercourses or bodies of water than Bareyo ?
Answer: 2 |

Table 5: Examples of question types attested in 𝕊ℙ𝕀ℂ𝔼 (Perez-Beltrachini et al., 2023). The utterance in gray is provided for ease of interpretation.