# OpenReview forum: "Conversational Semantic Parsing using Dynamic Context Graphs"
_EMNLP/2023/Conference — EMNLP 2023 Main_

### Official Review · Reviewer_4Uzo · 2023-08-06

**Typos Grammar Style And Presentation Improvements:** NA
**Soundness:** 3

**Excitement:**

3: Ambivalent: It has merits (e.g., it reports state-of-the-art results, the idea is nice), but there are key weaknesses (e.g., it describes incremental work), and it can significantly benefit from another round of revision. However, I won't object to accepting it if my co-reviewers champion it.

**Missing References:**

NA

**Paper Topic And Main Contributions:**

This paper proposes a semantic parser for KBQA. The model represents information about utterances and their context as KG subgraphs which are created dynamically and encoded using a graph neural network. Furthermore, the propose a context-dependent approach to type linking which is efficient and scalable.

**Questions For The Authors:**

See Reasons To Reject

**Reasons To Accept:**

(1) This paper is easy to understand
(2) The experiment results are strong

**Reasons To Reject:**

In the era of foundation model, it is beneficial to check the performance of ChatGPT.

**Reproducibility:**

3: Could reproduce the results with some difficulty. The settings of parameters are underspecified or subjectively determined; the training/evaluation data are not widely available.

**Reviewer Confidence:**

4: Quite sure. I tried to check the important points carefully. It's unlikely, though conceivable, that I missed something that should affect my ratings.

---

> ### Author Rebuttal · Authors · 2023-08-27
>
> Our work focuses on conversational semantic parsing over large-scale knowledge graphs. It is rather challenging to evaluate ChatGPT in this setting, for several reasons:
>
> a) data leakage, we cannot know which portions of our training/test data have been seen by ChatGPT;
>
> b) it is not clear how to encode the KG as context for ChatGPT;
>
> c) it is not obvious how to engineer prompts that can perform the reasoning required for the task and incorporate discourse context;
>
> d) evaluating ChatGPT's answers against our goldstandard would also require special treatment as ChatGPT's answers are typically a lot longer and exact match would in most cases fail.
>
> Such a study is not a simple baseline and but would rather merit a separate investigation which we leave to future work.

---

### Official Review · Reviewer_bYQ1 · 2023-08-07

**Soundness:** 4

**Excitement:**

4: Strong: This paper deepens the understanding of some phenomenon or lowers the barriers to an existing research direction.

**Paper Topic And Main Contributions:**

This paper investigates conversational semantic parsing over general purpose KG’s. They use dynamic context modeling, shows promising result in handling conversational features such as ellipses and co-references.  The area of open domain conversational Q&A using dynamic graphs is an under researched area, and the paper explores that.


**Reasons To Accept:**

•	The strengths of the paper are in the good description of the model, detailed analysis of the results, reasonable experiment structure and baselines.
•	Results are presented well with a detailed ablation study; The ablation analysis and case studies provide additional insight into the effectiveness of the approach.
•	Code and data are going to be released
•	It presents a novel approach with strong performance, and clear motivation.
•	This is a very well written paper, coherent and easy to understand.
•	Architecture of the model is well-defined in figure 2.
•	Overall, this is an effective paper with strong results

**Reasons To Reject:**

•Most of the improvement is coming from disambiguation based on popularity of an entity in training data.  More experiments are needed to see the generalizability of this approach on new dataset and unknown entities

**Reproducibility:**

4: Could mostly reproduce the results, but there may be some variation because of sample variance or minor variations in their interpretation of the protocol or method.

**Reviewer Confidence:**

4: Quite sure. I tried to check the important points carefully. It's unlikely, though conceivable, that I missed something that should affect my ratings.

---

> ### Author Rebuttal · Authors · 2023-08-27
>
> SPICE is the only dataset we are aware of that can serve as a testbed for conversational semantic parsing over KGs. We agree that evaluation of our method on other datasets with new entities is an avenue for future work, in particular since our results show that entity disambiguation plays a significant role for this task.

---

### Official Review · Reviewer_Qheq · 2023-08-11

**Soundness:** 3

**Excitement:**

3: Ambivalent: It has merits (e.g., it reports state-of-the-art results, the idea is nice), but there are key weaknesses (e.g., it describes incremental work), and it can significantly benefit from another round of revision. However, I won't object to accepting it if my co-reviewers champion it.

**Paper Topic And Main Contributions:**

This paper addresses the problem of semantic parsing by utilizing dynamic subgraphs. These subgraphs are built by utilizing the extracted entities during training. The paper runs their experiments on SPICE dataset and compares with one of the models that is introduced in the dataset paper. They provide an extended comparison between different variants of the models, and provide an analysis of different components. However, I believe the paper is missing a comparison with LasagneSP (Kacupaj et. al), which is the most similar with their work, and the main different is the global graph versus a dynamic graph that is introduced as novelty in this work. The results from LasagneSP also computed in SPICE paper (Perez-Beltrachini et al.) are not included in the results tables.

**Reasons To Accept:**

- An interesting approach which builds upon previous work and introduces dynamic context graph for semantic parsing
- Extended comparison and analysis with BertSP

**Reasons To Reject:**

- Analysis and comparison with similar model like LasagneSP is missing.
-  The model lacks novel design

**Reproducibility:**

5: Could easily reproduce the results.

**Reviewer Confidence:**

5: Positive that my evaluation is correct. I read the paper very carefully and I am very familiar with related work.

---

> ### Author Rebuttal · Authors · 2023-08-27
>
> We discuss the Lasagne model (Kacupaj et al., 2021) in the Related Work Section (line 176 onwards). However, their work is not directly comparable to ours because they assume a smaller, fixed ontology. This means that their model cannot be used (without modification) in a large-scale dynamic environment, like the one we are addressing in this paper. We will explain this in more detail. *Although the results are not directly comparable, we present below the performance of our model and Lasagne (as reported in Perez et al., 2023).*
>
> Our work outperforms Lasagne:
> | Model | F1 | EM
> |---|---|---|
> |Lasagne | 79.50 | 66.32 |
> | Ours | 81.28 | 66.85 |
>
> We are not aware of any previous work that integrates graph structural information with conversational semantic parsing. We are happy to accommodate any references the reviewer has in mind. As we discuss in Related Work, most previous approaches consider a non-conversational setting, while for those that are convesational the graphs are essentially table schemas which are significantly smaller and static. In contrast to Lasagne, our approach is executable and our knowledge graphs are created dynamically, i.e., the number of nodes varies per utterance.

---

### Meta-Review · Area_Chair_nu5Q · 2023-09-19

**Recommendation:** 5

**Metareview:**

This paper presents an approach to conversational semantic parsing over large knowledge graphs. The paper has unfortunately received rather superficial reviewing which could not be improved during the discussion period. In what follows, I summarize my own view (AC), departing from reviewers' evaluations.

The paper addresses an important task of conversational QA. This is a very relevant contribution -- and the findings are evaluated on a very recent (and already established) benchmark. The paper follows a principled approach, by introducing dynamic graph modeling ("dynamic context graph").

The paper presents a thorough and detailed evaluation. It shows a detailed comparison against SOTA and a very insightful analysis of the system's performance, going beyond simple numbers. The results suggest an improvement for both simple and complex questions.

Minor suggestion for the authors: Figure 1 is very instructive and does a good job at making your task clear -- however, the queries are impossible to understand and rather confusing. Would it be possible to either incorporate node names into the figure directly or provide some explanation on them a bit earlier than line 067?

---

### Decision · Program_Chairs · 2023-10-07

**Decision:**

Accept-Main

**Comment:**

This paper presents an approach to conversational semantic parsing over large knowledge graphs. The paper has unfortunately received rather superficial reviewing which could not be improved during the discussion period. In what follows, I summarize my own view (AC), departing from reviewers' evaluations.

The paper addresses an important task of conversational QA. This is a very relevant contribution -- and the findings are evaluated on a very recent (and already established) benchmark. The paper follows a principled approach, by introducing dynamic graph modeling ("dynamic context graph").

The paper presents a thorough and detailed evaluation. It shows a detailed comparison against SOTA and a very insightful analysis of the system's performance, going beyond simple numbers. The results suggest an improvement for both simple and complex questions.

Minor suggestion for the authors: Figure 1 is very instructive and does a good job at making your task clear -- however, the queries are impossible to understand and rather confusing. Would it be possible to either incorporate node names into the figure directly or provide some explanation on them a bit earlier than line 067?